# Optimising Hollow-Structured Silicon Nanoparticles for Lithium-Ion Batteries

**DOI:** 10.3390/ma16175884

**Published:** 2023-08-28

**Authors:** Chenghao Yue, Yao Liu, Shaoliang Guan, Alireza Fereydooni, Yuexi Zeng, Zhijie Wei, Yonggang Wang, Yimin Chao

**Affiliations:** 1School of Chemistry, University of East Anglia, Norwich Research Park, Norwich NR4 7TJ, UK; c.yue@uea.ac.uk (C.Y.); a.fereydoonisefiddashti@uea.ac.uk (A.F.); 2Department of Chemistry, Fudan University, Shanghai 200433, China; 3School of Chemistry, Cardiff University, Cardiff CF10 3AT, UK; shaoliang.guan@harwellxps.uk; 4Harwell XPS, Rutherford Appleton Laboratory, Didcot OX11 0FA, UK; 5Foshan Xianhu Laboratory, Advanced Energy Science and Technology Guangdong Laboratory, Foshan 528216, China

**Keywords:** silicon nanoparticles, Li-ion battery, anode, TiO_2_, capacity

## Abstract

Silicon has been proven to be one of the most promising anode materials for the next generation of lithium-ion batteries for application in batteries, the Si anode should have high capacity and must be industrially scalable. In this study, we designed and synthesised a hollow structure to meet these requirements. All the processes were carried out without special equipment. The Si nanoparticles that are commercially available were used as the core sealed inside a TiO_2_ shell, with rationally designed void space between the particles and shell. The Si@TiO_2_ were characterised using X-ray diffraction (XRD), X-ray photoelectron spectroscopy (XPS), and scanning electron microscopy (SEM). The optimised hollow-structured silicon nanoparticles, when used as the anode in a lithium-ion battery, exhibited a high reversible specific capacity over 630 mAhg^−1^, much higher than the 370 mAhg^−1^ from the commercial graphite anodes. This excellent electrochemical property of the nanoparticles could be attributed to their optimised phase and unique hollow nanostructure.

## 1. Introduction

The increasing demand for renewable energy, which stems from the harmful impact of global warming and the depletion of fossil fuels, has created a pressing need for reliable and efficient electric vehicles and energy storage systems [1,2,3,4]. Due to their high energy density, low self-discharge rate, and long cycle life, lithium-ion batteries (LIBs), in particular, have emerged as a leading technology for electrical/electronic facilities, information technology, aerospace, and electric vehicles (EVs) [5,6,7].

Nevertheless, further advancements in battery technology are necessary to meet the growing demand for energy storage. Using silicon as the anode material in LIBs provides ultra-high theoretical specific capacity up to 4200 mAhg^−1^, which is over 10 times higher than that of conventional graphite anodes (372 mAhg^−1^). Anode materials such as Li, Sn, and Al have also been extensively studied in this field [8,9,10]. Moreover, the availability of abundant Si resources, coupled with its low working potential and environmental friendliness, makes Si an ideal material for anodes [11].

Despite their high energy density, Si-based anodes have a well-documented tendency to expand during the lithiation/delithiation process, which can greatly influence the electrochemical performance of the battery. Furthermore, low electrical conductivity and slow ion diffusion kinetics are other reasons why Si anodes have not been widely commercialised [12,13]. This expansion tendency has posed a significant challenge, as it has the highest volume change of all the anode materials. This expansion leads to mechanical stress and strain on the surface of silicon materials, leading to cracking. Silicon nanoparticles with a size of *ca* 150 nm play a vital role, because cracking can be avoided when the size is less than this [14,15,16].

Nano-Si with high surface areas iswidely investigated due to the shortened pathways for lithium-ion diffusion and electron transport [17,18,19]. However, the high specific surface area of nanoscale Si also causes the formation of more SEI layers which could lead to a lower initial columbic efficiency. Combining Si with high conductivity materials can help in avoiding the direct contact between the nanoscale Si and electrolyte and release the volume change of Si during charging and discharging [20,21,22]. Thus, one potential solution to enhance the performance of Si is to design hollow-structured silicon nanoparticles, which could mitigate the expansion and cracking problems [1,23].

Here, we presented a hollow structure with silicon nanoparticles to improve the capacity of the anode and mitigate the volume expansion of silicon during cycling, where a titanium dioxide shell was designed to wrap the silicon nanoparticles. In this study, Ti^4+^ in ethanol was chosen to react with oxygen to form a TiO_2_ shell outside the silicon nanoparticles. Different thicknesses of TiO_2_ were explored to further stabilise and improve the performance of the SiNPs. The procedure shown in Figure 1 is a simple one-pot process and could be easily scaled up to an industry scale.

Owing to its low structural strain and non-toxicity, TiO_2_ has gained immense attention as a safe electrode material for lithium-ion batteries (LIBs). It has been reported to be a superb anode material due to its great chemical stability [24,25]. Titanium is used as part of anode materials with high specific capacity materials, such as Li and Al, to improve the stability of this kind of anode material [8,10].

## 2. Experimental Details

### 2.1. Material Fabrication

There were two samples consisting of silicon nanoparticles with different thickness of the TiO_2_ shell prepared with the following procedure.

An amount of 70 g glucose (>99.5%, Sigma-Aldrich, St. Louis, MO, USA) and 0.7 g silicon nanoparticles (60 nm, Thermo Fisher, Waltham, MA, USA) were mixed and sonicated in 70 mL DI water to obtain a dark brown suspension because the silicon nanoparticles were not dissolved in DI water. The dark brown suspension was transferred into a Teflon reactor and fixed into a stainless-steel autoclave, then the autoclave was put into the oven and kept at 180 °C for 8 h. After heating, the product was centrifuged three times at 10,000 rpm for 10 min to wash away the extra glucose. The obtained dark brown slurry after centrifuging was dried under vacuum for 4 h to obtain dry powder. After that, 20 mL of titanium isoproproxide (97%, Sigma-Aldrich) containing Ti^4+^ and 30 mL of ethanol were mixed and sonicated to make sure that Si@C was spread properly and then the mixture was aged for 12 h. The aged suspension was washed three times with the centrifuge to remove the extra Ti^4+^. After centrifuge washing, it was dried in the oven at 40 °C for 4 h to obtain the Si@C@TiO_2_. The resultant white powder was heated in the furnace with a 5 °C min^−1^ heating rate and kept at 550 °C for 2 h to obtain the final Si@TiO_2_.

Based on the battery performance achieved from the above recipe, the specific capacity was lower than expected and similar to TiO_2_.

This implied the TiO_2_ shell outside the silicon core was too thick. The amount of Ti^4+^ should be reduced to obtain a thinner shell. Then, the procedure was adjusted to 5 mL of titanium isoproproxide for ageing the Si@C.

### 2.2. Material Characterisation

The morphologies of the samples were investigated by scanning electron microscopy (SEM, Zeiss Gemini 300, ZEISS, Jena, Germany). The structure of the materials was characterised by an X-ray diffractometer (XRD, XtaLAB Synergy-S, JOEL, Tokyo, Japan) operated with a scanning speed of 5° min^−1^ from 20° to 90°. XPS Analysis was performed using a Thermo NEXSA XPS(Thermo Scientific, Waltham, USA) fitted with a monochromated Al kα X-ray source (1486.7 eV), a spherical sector analyser and 3 multichannel resistive plate, with 128 channel delay line detectors. All data was recorded at 19.2 W and an X-ray beam size of 400 µm × 200 µm. Survey scans were recorded at a pass energy of 200 eV, and high-resolution scans recorded at a pass energy of 40 eV. Electronic charge neutralization was achieved using a dual-beam low-energy electron/ion source (Thermo Scientific FG-03, Thermo Scientific, Waltham, USA). The ion gun current was 150 µA and the ion gun voltage was 45 V. All sample data was recorded at a pressure below 10-8 Torr and a room temperature of 294 K. Data was analysed using CasaXPS v2.3.20PR1.0o and calibrated with C1s peak at 284.8. eVDynamic light scattering (DLS) was employed to analyse the size of Si@TiO_2_ particles in dispersion and that was studied after it was dissolved in Ethonal.

### 2.3. Electrochemical Measurements

Half cells were employed to test the electrochemical performance of Si@TiO_2_ electrodes. The working electrode was composed of active material, conductive addition, and binder with a mass ratio of 8:1:1. As prepared Si@TiO_2_ was mixed with 10 wt.% carbon black and 10 wt.% PVDF in N-methyl pyrrolidone solution and the obtained slurry was coated on a copper foil. After drying in the vacuum oven, the electrodes (d = 16 mm) were prepared by punching machine.

The electrodes were assembled into half cells and full cells separately. An amount of 1 M LiPF_6_ dissolved in ethylene carbonate (EC) and dimethyl carbonate (DMC) (1:1 in volume) with 2% vinylene carbonate (VC) was chosen as the electrolyte. The coin type half cell (CR2016) was composed of an anode and pure lithium chip with a porous polypropylene separator between them. In the full cell, the lithium chip was replaced by LiCoO_2_ as the cathode, following the same procedure as in the half cell. Both the half cells and full cells were aged for 4 h before testing. Cyclic voltammetry (CV) measurement was carried out by using ChenHua CHI-660E (CH Instrument, Austin, TX, USA) electrochemical workstation under a scanning rate of 0.1 mVs^−1^ with a voltage window from 3 to 0.01 V (vs. Li^+^/Li). The cycling performance of the half cell and full cell were tested in constant current (CC) mode by Land CT3001 (LANHE, Wuhan, China) from 3 to 0.01 V (vs. Li^+^/Li) and from 4.2 to 3 V (vs. Li^+^/Li), respectively.

## 3. Results and Discussion

### 3.1. X-ray Photoelectron Spectroscopy 

XPS is a surface sensitive technique, measured to determine the elemental composition and chemical states. The XPS survey spectrum obtained from the raw silicon nanoparticles (SiNPs) is shown in Figure 2, where only Si2s, Si2p, and O1s peaks are visible. After heating the SiNPs and glucose with the initial recipe in the oven at 180 °C, a thick carbon shell was formed outside the SiNPs. The survey spectrum from this stage shows the elements of O, C, and Si in Figure 3: a very strong C1s peak and a weak signal from Si. The analysis shows 77.46% C1s and 21.55% O1s, while the percentage of silicon is 0.98%. The Si2p can be deconvoluted into three peaks corresponding to Si-C (102.93 eV), Si-O (104.22 eV), and Si-Si (99.22 eV), respectively. The data confirms there should be a thick shell outside the SiNPs, when we compare Si@C with the data from the raw SiNPs. The signal from silicon is weak, because it is not possible for the X-ray to penetrate more than 5–10 nm below the sample surface.

After Si@C is aged with titanium isopropoxide and heated to 550 °C in a furnace without air protection, a hollow-structured Si@TiO_2_ should be formed. The XPS spectra obtained from Si@TiO_2_ are shown in Figure 4, where the Ti peak is much stronger than the C1s peak because of the thick TiO_2_ shell wrapped outside the SiNPs. For the Ti2p spectrum shown in Figure 4c, the peaks centered at 458 eV and 463.7 eV belong to Ti2p3/2 and Ti2p1/2 orbitals of TiO_2_, respectively. In addition, the binding energy spacing between the two spin split orbits is 5.7 eV, indicating the state of Ti^4+^ in the anatase TiO_2_ [26]. However, the thick titanium shell is not doing well in regard to the expected improved battery performance. The high percentage of silicon within the sample may increase the specific capacity of the battery. The amount of titanium is therefore reduced to obtain a thinner shell outside the SiNPs. Figure 5 shows the XPS spectra after reducing the amount of titanium isoproproxide when ageing Si@C. The atomic percentage of silicon is 23.58%, which is over two times more than the atomic percentage of Ti, at 10.88%. This is evidence that the shell is thinner than in the previous sample. The high resolution XPS spectrum of Si2p is shown in Figure 5b, and the peaks located at 103.5 eV and 99.4 eV belong to Si-O and Si-Si, respectively, which are attributed to oxidation of the SiNPs. The thickness of the TiO_2_ shell can be measured by TEM, but TEM data is not included in this study.

### 3.2. X-ray Powder Diffraction

The observations made using XPS are also visible in the XRD spectrum; the XRD spectra obtained from Si@C and Si@C@TiO_2_ are shown in Appendix A. The XRD spectrum obtained from Si@TiO_2_ with a lower amount of Ti is shown in Figure 6. Si 111, Si 220, Si 400, and TiO_2_ all are visible in the XRD pattern, which could prove that the SiNPs are wrapped by TiO_2_ with the data observed from the XPS spectra.

### 3.3. Scanning Electron Microscope

SEM was utilised to determine the size and shape of Si@TiO_2_. The SEM image of raw SiNPs is shown in Figure 7a. After the 60 nm SiNOs were mixed with DI water and glucose, and the mixture was heated to 180 °C for 12 h, there was a thick carbon shell outside the SiNPs. This was also proven by the XPS spectra of Si@C in Figure 3, where the percentage of C was 77.46%. The shell was so thick that the XPS could only detect a very weak signal from the SiNPs inside. The SEM image, as shown in Figure 7b, shows the size ca 500 nm, which was much larger than the raw material.

Si@C@TiO_2_ was formed after ageing Si@C with titanium isoproproxide. From the SEM image, shown in Figure 7c, one can observe that the size is similar to that of Si@C. Finally, the Si@C@TiO_2_ was heated to 550 °C under the ambient atomsphere without any protection. The solution of titanium isoprorpoxide contained Ti^4+^, which reacted with oxygen to form a TiO_2_ shell. After burning away the carbon between the SiNPs and TiO_2_, which is proved by the observation by the XPS spectra from Si@TiO_2_ shown in Figure 4, the size was similar compared to the previous Si@C or Si@C@TiO_2_, but many anatase shells also merged with each other, thus forming connected yolk–shell Si@TiO_2_ clusters, like biological tissue that contains many adjacent cells [27] (see Figure 7d).

The SEM images can demonstrate whether the TiO_2_ shell has wrapped multiple SiNPs together. If more than one nanoparticle is wrapped by the TiO_2_ shell, the size and shape would be changed. From the images for the Si@TiO_2_, multiple sizes of the nanoparticles were obtained because not only one SiNPs is wrapped by the TiO_2_ shell, since the size of the raw SiNPs used for the synthesis is only 60 nm.

### 3.4. Dynamic Light Scattering

DLS measurement is a powerful technique used to study the size distribution of inorganic particles. Si@TiO_2_ with a lower amount of Ti was dispersed in ethanol and sonicated before measurement. Figure 8 shows that most of the Si@TiO_2_ particles under investigation fell within the size range from 200 to 800 nm, as shown in Figure 8. There were some large size particles over 800 nm, which should be anatase shells merged with each other, forming connected yolk–shell Si@TiO_2_.

DLS measurement can provide the key information of the particle size distribution. Combined with the SEM image, the DLS data confirms the major size of Si@TiO_2_ is around 500 nm, and the size can be over 1 µm if more than one SiNPs is wrapped by the TiO_2_ shell.

### 3.5. Half-Cell and Full-Cell Cycling Test

Before the adjustment of the amount of titanium isoprorpoxide, the battery performance is shown in Figure 9a, when assembling battery with the thicker titanium dioxide shell encapsulated the SiNPs. An amount of 1 mg Si@TiO_2_ was loaded onto the electrode and was assembled with a pure lithium chip in CR2016, and tested at 50 mAg^−1^ in the voltage range 0–3.0 V. The discharge capacities of the initial three cycles were 341, 235, and 233 mAhg^−1^, respectively. The discharge capacity quickly dropped to 119 mAhg^−1^ after 15 cycles. The CV curve is illustratedin Figure 9b, which displays the reduction–oxidation reaction of the silicon-based anode during the first charging and discharging cycle within the voltage range 0–3.0 V at a scan rate of 0.1 mV s^−1^. There was a primary reduction peak at about 0.6 V, which mainly corresponded to the silicon and a sharp peak around 1.5 V which belonged to the TiO_2_. The observed oxidation peaks at 0.7 V and 1.4 V should be attributed to the lithium insertion into the silicon and TiO_2_. The TiO_2_ shell was too thick for the second charge–discharge cycles. The thick TiO_2_ shell made the main contribution to the redox reaction, while the characteristic peaks of the deep silicon core were not so obvious.

The specific capacity of the higher amount of titanium was around 200 mAhg^−1^; even in the first cycle it was 340 mAhg^−1^, which was far away from the theoretical specific capacity of silicon but similar to the performance of titanium. The working voltage was around 1.8 V which was quite high when compared with silicon. Combining this data with the CV, the previous XPS and XRD data, the titanium dioxide shell was too thick, so the silicon core did not contribute to the reaction during the charging and discharging cycles.

The Si@TiO_2_ with the small quantities of Ti electrode was assembled in a half cell with a pure lithium chip, the same as the high amount of Ti. After reducing the amount of titanium, the capacity of Si@TiO_2_ was improved significantly, as the 1st, 5th, 10th, 20th, 30th, and 50th cycle show in Figure 10. In the first and fifth cycle discharge voltage curve, a small plateau around 0.4 V may have been due to the formation of solid electrolyte interface (SEI) layers, and the long flat plateau around 0.15 V corresponded to the Li^+^ insertion into Si. The specific capacity of the first cycle was up to 630 mAhg^−1^, which was almost double the value from the current commercial graphite used in lithium-ion batteries, which is around 370 [28]. The coulombic efficiency of both Si@TiO_2_ with low and high amount of Ti was around 100% in the half cell (see Appendix A). In addition, the stability was much better than the heavy titanium sample. After 50 cycles, the specific capacity was still maintained at 440 mAhg^−1^, which was approximately 70% of the value in the initial cycle.

A half-cell battery is not suitable to be used in real daily life; it is just for research purposes to obtain data from the targeted materials. It is therefore necessary to assemble and test a full-cell battery to further investigate the feasibility of the Si@TiO_2_ electrode for practical application. A Li-ion full cell was assembled with Si@TiO_2_ as the anode and LiCoO_2_ as the cathode. LiCoO_2_ is one of the earliest commercial cathodes; presently, it is still the flagship commercialised cathode in the lithium-ion battery market, mostly for electronic consumer products. It shows advantages such as high theoretical capacity up to 274 mAhg^−1^, excellent rate capability, etc. [29,30]. After balancing the merits between energy density and reversibility, LiCoO_2_ was selected as the cathode material [31]. Figure 11 shows the charge/discharge voltage profile of the Si@TiO_2_/LiCoO_2_ full cell tested at 0.1 C (27.4 mAg^−1^, theory specific capacity of LiCoO_2_ is 274 mAhg^−1^) in the voltage range of 3–4.2 V. The discharge capacities (calculated based on the mass of the anode material) at the initial three cycles were 304, 299, and 294 mAhg^−1^, respectively. The charge/discharge capacity decreased from 304 to 277 mAhg^−1^ at the initial 10 cycles. The coulombic efficiency is shown in Appendix A. The high discharge capacity indicated the Si@TiO_2_ has great potential for commercial application.

## 4. Conclusions

In conclusion, our study focused on designing a hollow-structured silicon nanoparticle anode for lithium-ion batteries. The results indicate that this approach can significantly improve the performance of the battery, up to 630 mAhg^−1^ in half cell and up to 300 mAhg^−1^ in full cell. The hollow structure of the SiNPs provides ample space for the volume expansion that occurs during cycling, thereby reducing the risk of particle fracture and increasing the overall stability of the battery. Adjusting the amount of Ti can improve the performance of the Si@TiO_2_ electrode. Furthermore, the high specific capacity of silicon, combined with the hollow structure, leads to enhanced energy storage and improved battery performance.

## Figures and Tables

**Figure 1 materials-16-05884-f001:**
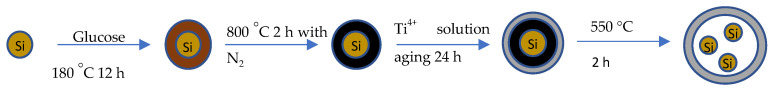
Synthesis procedure of hollow-structured Si@TiO_2_.

**Figure 2 materials-16-05884-f002:**
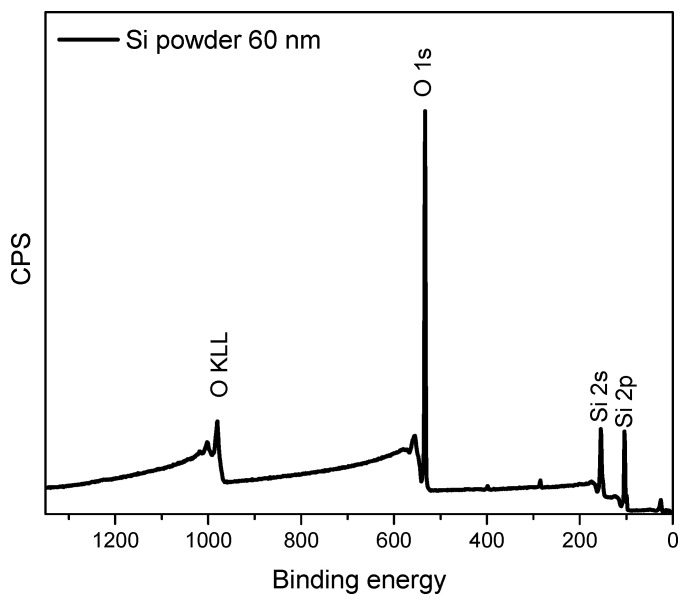
XPS survey spectrum obtained from Si powder (Step: 1 eV).

**Figure 3 materials-16-05884-f003:**
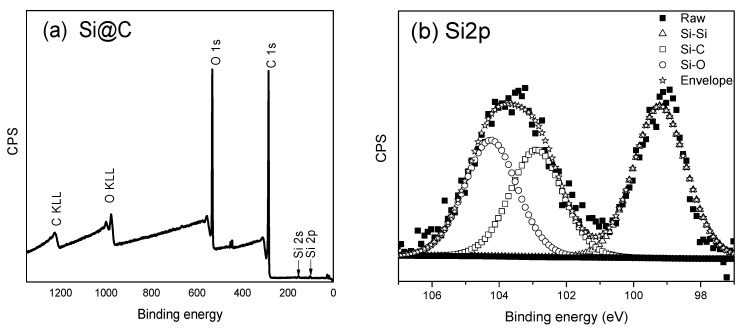
XPS spectra obtained from Si@C (**a**) XPS survey spectrum from Si@C with 1 eV step (**b**) high-resolution spectrum showing Si2p region with 0.1 eV step (**c**) high-resolution spectrum showing of O1s with 0.1 eV step (**d**) high-resolution spectrum showing C1s region with 0.1 eV step.

**Figure 4 materials-16-05884-f004:**
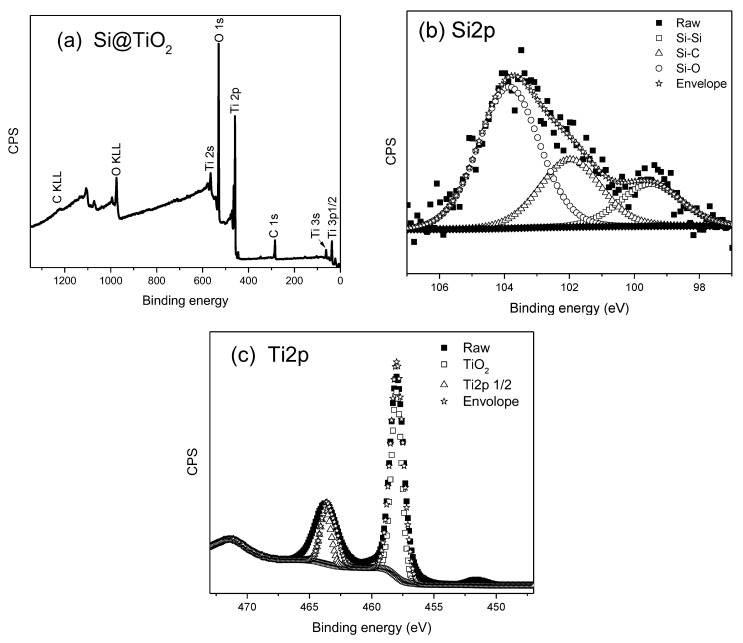
XPS spectrum obtained from Si@TiO_2_ (**a**) XPS survey spectrum with 1 eV step (**b**) high-resolution spectrum showing Si2p region with 0.1 eV step (**c**) high-resolution spectrum showing Ti2p3/2 and Ti2p1/2 region with 0.1 eV step.

**Figure 5 materials-16-05884-f005:**
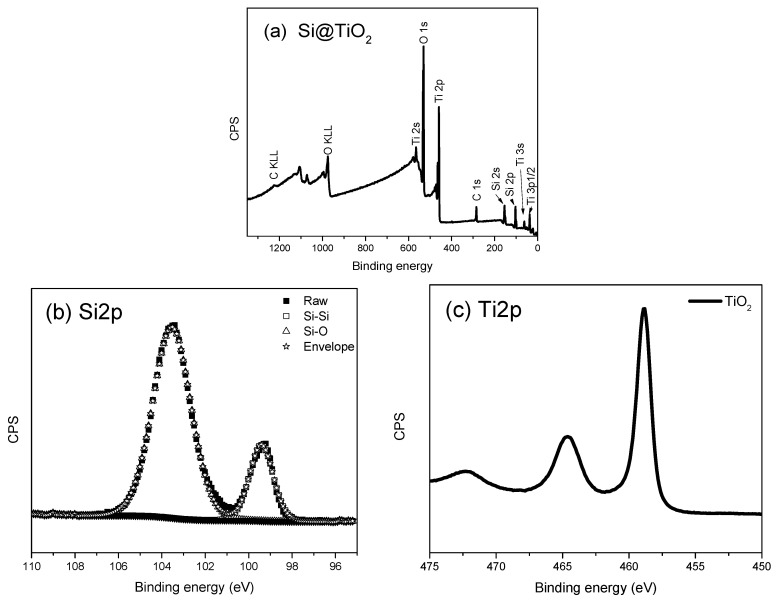
XPS spectrum obtained from Si@TiO_2_ with less amount of titanium (**a**) XPS survey spectrum with less amount of titanium (**b**) high-resolution spectrum showing Si2p region with 0.1 eV step (**c**) high-resolution spectrum showing Ti2p3/2 region with 0.1 eV step.

**Figure 6 materials-16-05884-f006:**
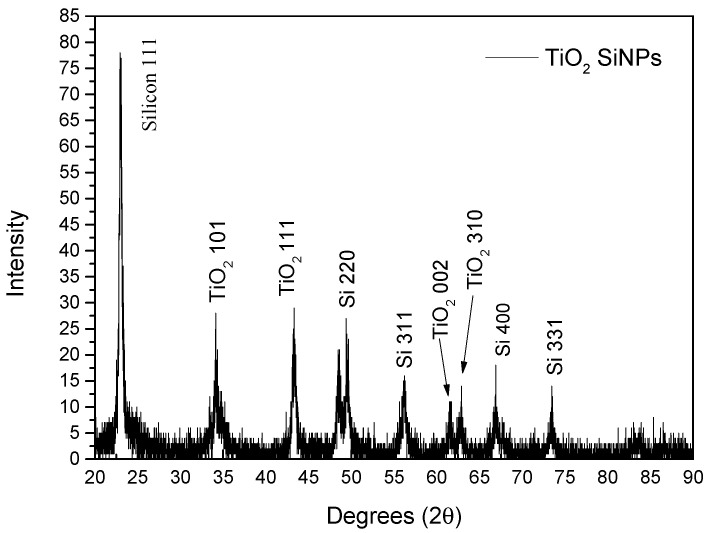
XRD spectrum obtained from Si@TiO_2_.

**Figure 7 materials-16-05884-f007:**
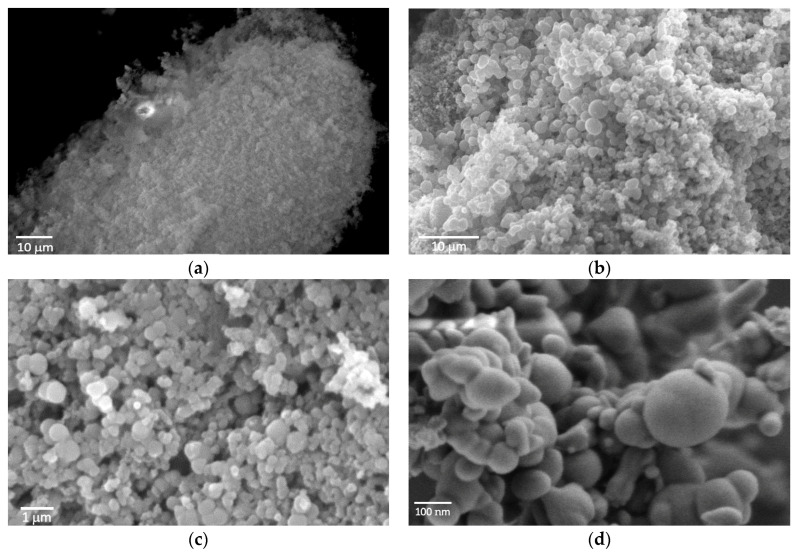
SEM image of (**a**) raw 60 nm Si nanoparticles used for the synthesis, (**b**) Si@C after burning glucose and Si nanoparticles in the oven at 180 °C, (**c**) Si@C@TiO_2_ after ageing the Si@C with titanium isoproproxide, (**d**) Si@TiO_2_ after removing C in furnace.

**Figure 8 materials-16-05884-f008:**
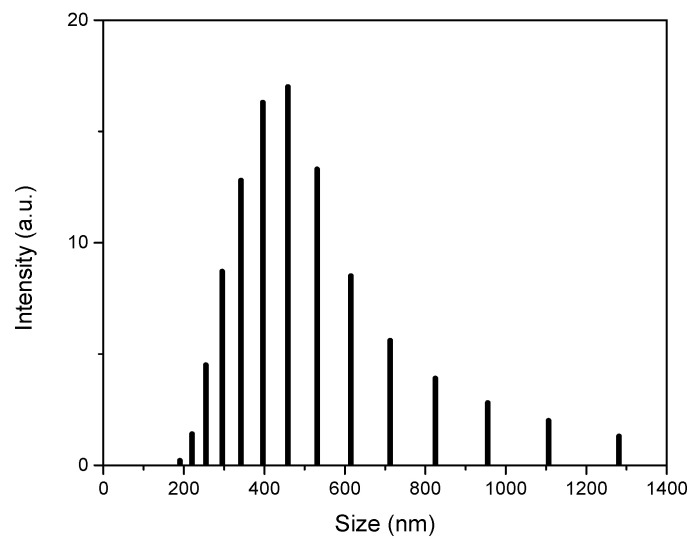
DLS measurement of Si@TiO_2_.

**Figure 9 materials-16-05884-f009:**
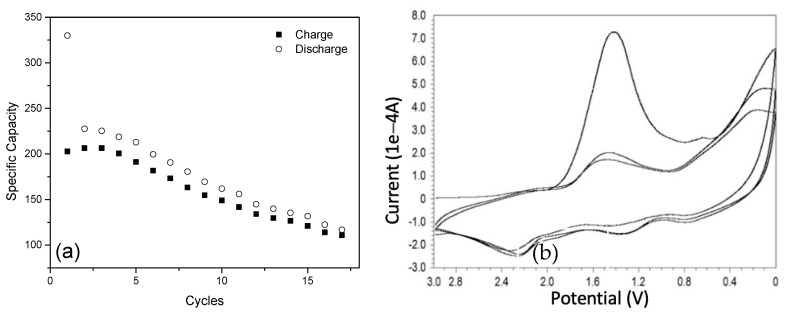
Battery performance of Si@TiO_2_ (high amount of Ti): (**a**) Cycling data from Si@TiO_2_, and (**b**) CV curve of Si@TiO_2_.

**Figure 10 materials-16-05884-f010:**
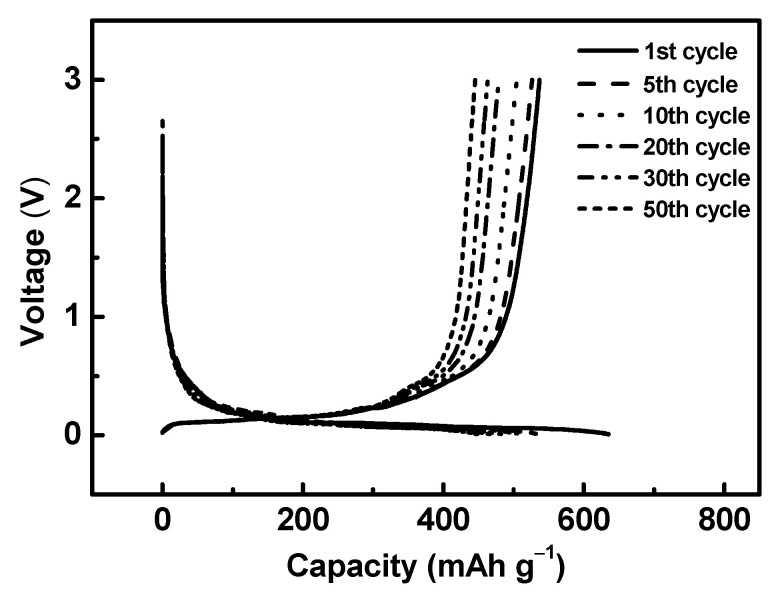
Specific capacity from Si@TiO_2_ (small quantities) and cycling data from Si@TiO_2_ (1, 5, 10, 20, 30, and 50 cycles).

**Figure 11 materials-16-05884-f011:**
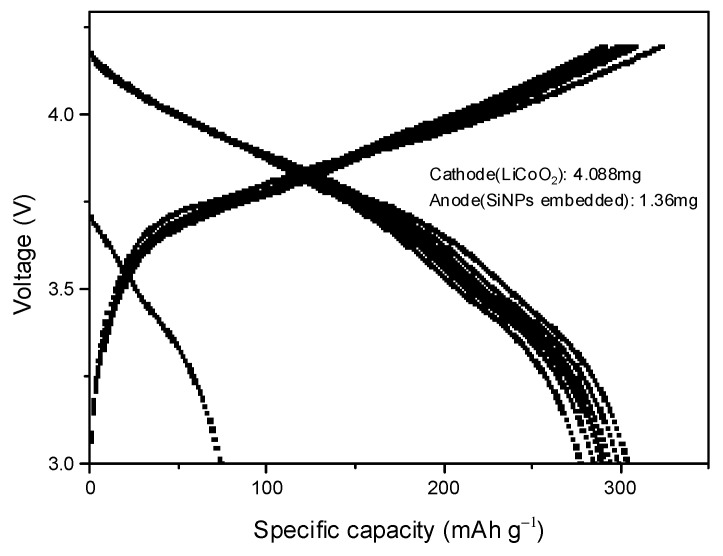
Full-cell cycling data from Si@TiO_2_ (anode mass loaded: 1.36 mg, cathode mass loaded: 4.088 mg).

## Data Availability

The data presented in this study may be available on reasonable request from the first or corresponding author.

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
