# Peer review of "Optimising Hollow-Structured Silicon Nanoparticles for Lithium-Ion Batteries"

_materials, 2023, doi:10.3390/ma16175884_

Round 1

Reviewer 1 Report

I have reviewed the manuscript titled “Optimising hollow structured silicon nanoparticles for lithium-ion batteries”. The manuscript presents an investigation into the performance of silicon nanoparticles with titanium dioxide shells as anodes in lithium-ion batteries. The authors have explored the effects of varying titanium content and shell thickness on specific capacity, cycle performance, and full-cell application. The research's novelty and potential implications for energy storage technologies are noteworthy. Based on my evaluation, I believe that the manuscript possesses substantial merit and makes a significant contribution to the field. However, it requires some revisions to address the following concerns.

1. The discussion regarding the thickness of the titanium dioxide shell affecting the silicon core's contribution to the reaction is intriguing. Elaboration on the methods used to determine the shell thickness, as well as more detailed analysis of how it impacts the silicon's ability to contribute to the reaction, would enhance the manuscript.

2. A comprehensive analysis of the cycle performance, including statistical methods to show the significance of the changes observed, would strengthen the findings.

3. The transition from half-cell to full-cell testing is mentioned, but additional details on this process and the choice of cathode materials would offer a more in-depth perspective.

4. Contextualizing the findings with existing technologies and materials would enhance the manuscript's value and would provide a clearer context for the improvements offered by the Si@TiO2 system.

5. The abstract and Conclusion are somewhat scanty because the salient results cited therein are entirely qualitative. It is necessary to include some important numerical results.

6. Ensure consistency in units (e.g., mAhg-1 vs. mAh/g).

Minor proofreading is required.

Author Response

  1. The discussion regarding the thickness of the titanium dioxide shell affecting the silicon core's contribution to the reaction is intriguing. Elaboration on the methods used to determine the shell thickness, as well as more detailed analysis of how it impacts the silicon's ability to contribute to the reaction, would enhance the manuscript.

Author’s response:

HRTEM would be possible to determine the shell thickness, but that is out of this work, and we will explore this feature in the future work.

  1. A comprehensive analysis of the cycle performance, including statistical methods to show the significance of the changes observed, would strengthen the findings.

Author’s response:

More information of half-cell and full-cell cycle performance has been added to the supporting information (SI), see Figure S3 and S4.

  1. The transition from half-cell to full-cell testing is mentioned, but additional details on this process and the choice of cathode materials would offer a more in-depth perspective.

Author’s response:

More background information of cathode has been added to the main text. It reads as:

A Li-ion full cell was assembled with Si@TiO­2 as anode and the LiCoO2 as cathode. LiCoO2 is one of the earliest commercial cathodes, presently, it is still the flagship commercialized cathode in the lithium-ion battery market, mostly for electronic consumer products. It shows advantages such as high theoretical capacity up to 274 mAhg-1, excellent rate capability, etc.[27, 28] After balancing merits between energy density and reversibility, LiCoO2 is selected as the cathode materials.[29]

  1. Contextualizing the findings with existing technologies and materials would enhance the manuscript's value and would provide a clearer context for the improvements offered by the Si@TiO2 system.

Author’s response:

A comparison with the common commercial anode, graphite, was made in the main text, our material can improve the capacity from 370 mAhg-1 which is almost the theoretic specific capacity of graphite to 630 mAhg-1.

It reads as

The specific capacity of the first cycle is up to 630 mAhg-1, which is almost double the value from the current commercial graphite used in lithium-ion batteries, which is around 370. [26] The coulombic efficiency of both Si@TiO2 with low and high amount of Ti is around 100% in half cell, see Figure S3 in supporting information.

  1. The abstract and Conclusion are somewhat scanty because the salient results cited therein are entirely qualitative. It is necessary to include some important numerical results.

Author’s response:

Some numerical results added.

  1. Ensure consistency in units (e.g., mAhg-1vs. mAh/g).

Author’s response:

All corrected now.

Reviewer 2 Report

The manuscript material is in demand. But there are two questions

1.  Insufficient introduction. There is no task setting. Why is this direction necessary to explore?

2. When there is chemistry in solutions, there must be proof of the pure product, including X-ray diffraction analysis.

No obvious errors

Author Response

The manuscript material is in demand. But there are two questions

  1. Insufficient introduction. There is no task setting. Why is this direction necessary to explore?

Author's response:

The following sentence has been added in introduction section: ‘Different thicknesses of TiO2 were explored to further stabilise and improve the performance of SiNPs.’

  1. When there is chemistry in solutions, there must be proof of the pure product, including X-ray diffraction analysis.

Author's response:

If we understand this correctly, the referee is suggesting that we should add X-ray diffraction analysis of each step of the whole process. XRD spectra of Si@C and Si@C@TiO2 have been added to the supporting information (SI), see Figure S1 and S2.

Round 2

Reviewer 1 Report

The authors have adequately addressed the concerns I raised in my initial review of the manuscript titled "Optimising hollow structured silicon nanoparticles for lithium-ion batteries". The revised manuscript is much improved, and I recommend it for publication.

Minor proofreading is required.